# Explicit Time-Domain Approach for Random Vibration Analysis of Jacket Platforms Subjected to Wave Loads

**Wei Lin [1] , Cheng Su [1,2,* and Youhong Tang [3]**

1   School of Civil Engineering and Transportation, South China University of Technology, Guangzhou 510640, China; wlin@scut.edu.cn
2   State Key Laboratory of Subtropical Building Science, South China University of Technology, Guangzhou 510640, China
3   College of Science and Engineering, Flinders University, South Australia 5042, Australia; youhong.tang@flinders.edu.au
*   Correspondence: cvchsu@scut.edu.cn; Tel.: +86-20-8711-1636

**Abstract:** This paper is devoted to the random vibration analysis of jacket platforms under wave loads using the explicit time-domain approach. The Morison equation is first used to obtain the nonlinear random wave loads, which are discretized into random loading vectors at a series of time instants. The Newmark-$\beta$ integration scheme is then employed to construct the explicit expressions for dynamic responses of jacket platforms in terms of the random vectors at different time instants. On this basis, Monte Carlo simulation can further be conducted at high efficiency, which not only provides the statistical moments of the random responses, but also gives the mean peak values of responses. Compared with the traditional power spectrum method, nonlinear wave loads can be readily taken into consideration in the present approach rather than using the equivalent linearized Morison equation. Compared with the traditional Monte Carlo simulation, the response statistics can be obtained through the direct use of the explicit expressions of dynamic responses rather than repeatedly solving the equation of motion. An engineering example is analyzed to illustrate the accuracy and efficiency of the present approach.

**Keywords:** jacket platform; wave load; random vibration; explicit time-domain method; Monte Carlo simulation

## 1. Introduction

Jacket platforms are the most widely used fixed offshore platforms in oil and gas exploitation, working at optimal operational water depths ranging from 100 to 200 m. The wave loads often dominate the environmental effects on jacket platforms. In particular, due to the increase in operational water depth, the natural frequencies of jacket platforms decrease and come close to the predominant frequencies of the wave energy, and the responses of jacket platforms are almost totally dependent on wave loads. To effectively support the topside facilities, the jacket structure is always fabricated as a welded tubular space frame consisting of vertical or battered legs strengthened by lateral bracing systems. Because the components of the jacket are slender cylinders with small diameters compared to the wavelength, the Morison equation can be applied to evaluate wave loads on jacket structures [1–6].

Wave loads are random in nature, and the responses of jacket platforms subjected to wave loads are random processes. Therefore, it is necessary to conduct random vibration analysis of jacket platforms to obtain the response statistics. Currently, the power spectrum method (PSM) is the dominant approach for random vibration analysis of jacket structures under random wave loads [7,8], in which the power

spectra of random wave loads need to be determined first. However, the wave loads represented by the Morison equation are nonlinear functions of wave-particle velocities. To determine the power spectra of wave loads in terms of the power spectra of wave-particle velocity, it is required to conduct the statistical linearization of the nonlinear Morison equation, in which the mean square of the error between the nonlinear and the linearized form of the Morison equation is minimized [9,10]. It has been observed that the above statistical linearization technique will significantly underestimate the mean peak values of the dominating wave drags for high sea conditions [11]. In view of this, an alternative linearization principle can be adopted, in which the mean peak wave load in the nonlinear case is equated with that in the linear case [12]. Nevertheless, certain assumptions regarding the probability distribution of the amplitude of wave-particle velocity are required in the linearized process when using the new principle, which inevitably influences the accuracy of the mean peak values of structural responses.

To fully consider the nonlinear wave loads represented by the nonlinear Morison equation, the random vibration of jacket platforms should be conducted in the time domain. In this paper, the random response of a jacket platform subjected to wave loads is analyzed with the explicit time-domain method (ETDM), which was originally proposed for random vibration analysis of building and bridge structures subjected to random seismic excitations [13–16]. Instead of using the equivalent linearized Morison equation, the nonlinear random wave loads can be directly taken into consideration in ETDM without any difficulties. The Newmark-$\beta$ integration scheme is first employed to establish the explicit expressions of dynamic responses of jacket platforms in terms of nonlinear random wave loads at different time instants. Using the explicit formulations of structural responses, Monte Carlo simulation (MCS) can then be conducted at high efficiency for achieving the statistical moments and the mean peak values of random responses [17], in which no repetitive solutions to the equation of motion are required for a large number of time-history analyses of the structure under different samples of wave loads. An engineering example involving a jacket platform with 11,688 degrees of freedom (DOFs) is analyzed to illustrate the accuracy and efficiency of the present approach.

## 2. Determination of Wave Loads with Nonlinear Morison Equation

Assume that the inclined cylinder $C_1C_2$, as shown in Figures 1–3, is a typical component of a jacket platform submerged in sea water. In these figures, the $x$-direction refers to the direction of wave propagation, and the $z$-direction refers to the vertical direction with the origin O located at the static water level. The inclined angles of cylinder $C_1C_2$ with respect to the $x$-, $y$-, and $z$-direction are denoted as $\varphi_x$, $\varphi_y$, and $\varphi_z$, respectively, as also presented in Figures 1–3. For an arbitrary point $P(x, y, z)$ on cylinder $C_1C_2$, the wave-particle velocities along the $x$- and $z$-direction are denoted as $v_x$ and $v_z$, respectively, for a two-dimensional swell model, and the corresponding wave-particle accelerations are denoted as $\dot{v}_x$ and $\dot{v}_z$, respectively, as shown in Figure 1. For the same point $P(x, y, z)$ on cylinder $C_1C_2$, the structural vibration velocities along the $x$-, $y$-, and $z$-direction are denoted as $\dot{u}_x$, $\dot{u}_y$, and $\dot{u}_z$, respectively, and the corresponding structural vibration accelerations are denoted as $\ddot{u}_x$, $\ddot{u}_y$, and $\ddot{u}_z$, respectively, as shown in Figure 2. The cylinder $C_1C_2$ is subjected to wave loads, and the distributed wave loads acting at point $P(x, y, z)$ along the $x$-, $y$-, and $z$-direction are denoted as $f_x$, $f_y$, and $f_z$, respectively, as shown in Figure 3.

Using the nonlinear Morison equation and based on the quantities defined in Figures 1–3, the total distributed wave loads can be derived as follows [18,19]:

$$\left\{ \begin{array}{c} f_x \\ f_y \\ f_z \end{array} \right\} = -m_A \left\{ \begin{array}{c} \ddot{u}_x \\ \ddot{u}_y \\ \ddot{u}_z \end{array} \right\} - c_H \left\{ \begin{array}{c} \dot{u}_x \\ \dot{u}_y \\ \dot{u}_z \end{array} \right\} + K_M \left\{ \begin{array}{c} \dot{v}_{Nx} \\ \dot{v}_{Ny} \\ \dot{v}_{Nz} \end{array} \right\} + K_D |v_N| \left\{ \begin{array}{c} v_{Nx} \\ v_{Ny} \\ v_{Nz} \end{array} \right\} \tag{1}$$

where $K_M = \frac{1}{4}C_M \rho \pi D^2$ and $K_D = \frac{1}{2}C_D \rho D$; $\rho$ is the water density and $D$ is the diameter of the cylinder; $C_M$ is the inertia coefficient and $C_D$ is the drag coefficient; $m_A = \frac{1}{4}C_A \rho \pi D^2$ is the added mass with

$C_A = C_M - 1$ being the added mass coefficient of the cylinder; $m$ is the mass per unit length of the cylinder; $c_H$ is the distributed hydrodynamic damping matrix; and $v_{Nx}$, $v_{Ny}$, and $v_{Nz}$ are the three components of the wave-particle velocity normal to the cylinder, $\boldsymbol{v}_N$, which is defined as:

$$\boldsymbol{v}_N = \begin{Bmatrix} v_{Nx} \\ v_{Ny} \\ v_{Nz} \end{Bmatrix} = \begin{bmatrix} 1 - \cos^2 \varphi_x & -\cos\varphi_x \cos\varphi_z \\ -\cos\varphi_x \cos\varphi_y & -\cos\varphi_y \cos\varphi_z \\ -\cos\varphi_x \cos\varphi_z & 1 - \cos^2\varphi_z \end{bmatrix} \begin{Bmatrix} v_x \\ v_z \end{Bmatrix} \tag{2}$$

with $|\boldsymbol{v}_N| = \sqrt{v_{Nx}^2 + v_{Ny}^2 + v_{Nz}^2}$.

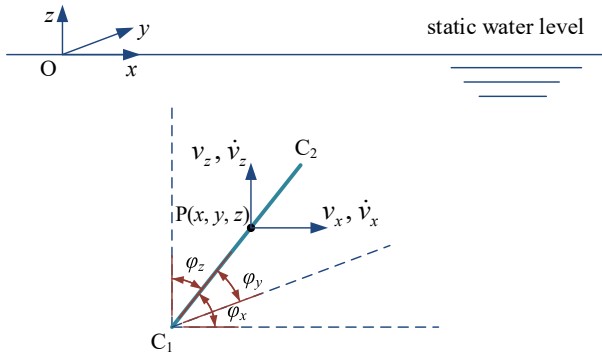

**Figure 1.** Wave-particle velocities and accelerations at point $P(x, y, z)$ on cylinder $C_1 C_2$.

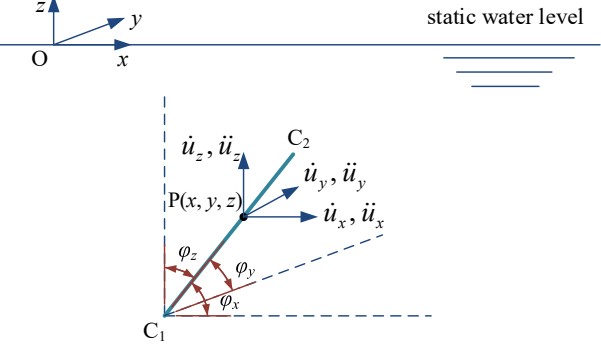

**Figure 2.** Structural vibration velocities and accelerations at point $P(x, y, z)$ on cylinder $C_1 C_2$.

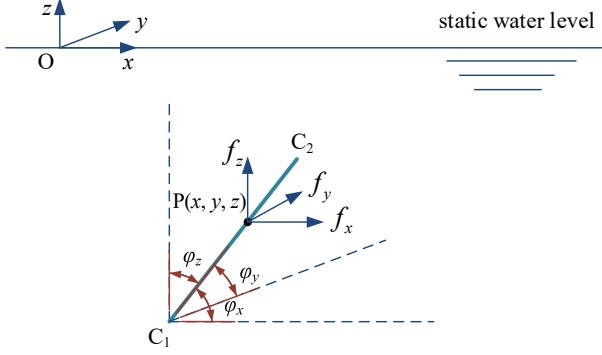

**Figure 3.** Distributed wave loads acting at point $P(x, y, z)$ on cylinder $C_1 C_2$.

Equation (1) can be rewritten in a more compact form as:

$$f_T = f_A + f_H + f_I + f_D \tag{3}$$

where $f_T = \begin{bmatrix} f_x & f_y & f_z \end{bmatrix}^T$ is the total distributed wave loading vector, in which the superscript T denotes the matrix transposition; and $f_A, f_H, f_I$, and $f_D$ represent the distributed added inertial force, hydrodynamic damping force, incident force and drag force vector, respectively, which are expressed as:

$$f_A = -m_A \begin{Bmatrix} \ddot{u}_x \\ \ddot{u}_y \\ \ddot{u}_z \end{Bmatrix}, \quad f_H = -c_H \begin{Bmatrix} \dot{u}_x \\ \dot{u}_y \\ \dot{u}_z \end{Bmatrix}, \quad f_I = K_M \begin{Bmatrix} \dot{v}_{Nx} \\ \dot{v}_{Ny} \\ \dot{v}_{Nz} \end{Bmatrix}, \quad f_D = K_D |v_N| \begin{Bmatrix} v_{Nx} \\ v_{Ny} \\ v_{Nz} \end{Bmatrix} \tag{4}$$

Note that in Equation (4) the drag forces in $f_D$ are nonlinear functions of the wave-particle velocities, and therefore the time-domain analysis, are required to consider the nonlinear drag forces.

## 3. Explicit Formulation of Dynamic Responses

### 3.1. Equation of Motion

The equation of motion for a jacket platform subjected to wave loads can be expressed as

$$M\ddot{U} + C\dot{U} + KU = LF_T \tag{5}$$

where $M$, $C$, and $K$ are the mass, damping, and stiffness matrix of the structure, respectively; $U$, $\dot{U}$, and $\ddot{U}$ denote the nodal displacement, velocity, and acceleration vector of the structure, respectively; and $L$ is the orientation matrix of the total concentrated wave loading vector $F_T$, which can be determined through the distributed wave loads shown in Equation (3) and can be expressed as:

$$F_T = F_A + F_H + F_I + F_D \tag{6}$$

where $F_A$, $F_H$, $F_I$, and $F_D$ are the concentrated added inertial force, hydrodynamic damping force, incident force, and drag force vector, respectively.

From the expressions of the distributed added inertial forces and hydrodynamic damping forces shown in Equation (4), $LF_A$ and $LF_H$ can be expressed in the following forms:

$$LF_A = -M_A\ddot{U}, \quad LF_H = -C_H\dot{U} \tag{7}$$

where $M_A$ and $C_H$ are the added mass matrix and the hydrodynamic damping matrix, respectively.

Substituting Equations (6) and (7) into Equation (5) yields:

$$(M + M_A)\ddot{U} + (C + C_H)\dot{U} + KU = L(F_I + F_D) = LF \tag{8}$$

where $F$ is the concentrated wave loading vector dependent on the wave-particle velocities and accelerations, which can be determined through the distributed incident forces and drag forces shown in Equation (4).

### 3.2. Explicit Time-Domain Expressions of Structural Responses

Define the state vector as $V = \begin{bmatrix} U^T & \dot{U}^T \end{bmatrix}^T$. The recurrence formula for the state vector can be expressed as [13]:

$$V_i = TV_{i-1} + Q_1 F_{i-1} + Q_2 F_i \quad (1 \le i \le n) \tag{9}$$

where $n$ is the number of time steps for time-history analysis; $V_i = V(t_i)$, $V_{i-1} = V(t_{i-1})$, $F_i = F(t_i)$, and $F_{i-1} = F(t_{i-1})$, in which $t_i = i\Delta t$, $t_{i-1} = (i-1)\Delta t$, and $\Delta t$ is the time step; and $T$, $Q_1$, and $Q_2$ are

the coefficient matrices derived through the Newmark-$\beta$ integration scheme [14–16], which can be expressed as:

$$
\begin{cases}
T = \begin{bmatrix} H_{11} & H_{12} \\ H_{21} & H_{22} \end{bmatrix}, \quad Q_1 = \begin{bmatrix} R_1 \\ R_3 \end{bmatrix} L, \quad Q_2 = \begin{bmatrix} R_2 \\ R_4 \end{bmatrix} L \\
H_{11} = \hat{K}^{-1}\left[S_1 - S_3(M+M_A)^{-1}K\right], \quad H_{12} = \hat{K}^{-1}\left[S_2 - S_3(M+M_A)^{-1}(C+C_H)\right] \\
H_{21} = a_3(H_{11} - I) + a_5(M+M_A)^{-1}K, \quad H_{22} = a_3 H_{12} - a_4 I + a_5(M+M_A)^{-1}(C+C_H) \\
R_1 = \hat{K}^{-1}S_3(M+M_A)^{-1}, \quad R_2 = \hat{K}^{-1}, \quad R_3 = a_3 R_1 - a_5(M+M_A)^{-1}, \quad R_4 = a_3 R_2 \\
\hat{K} = K + a_0(M+M_A) + a_3(C+C_H) \\
S_1 = a_0(M+M_A) + a_3(C+C_H), \quad S_2 = a_1(M+M_A) + a_4(C+C_H), \\
S_3 = a_2(M+M_A) + a_5(C+C_H) \\
a_0 = \frac{1}{\beta \Delta t^2}, \ a_1 = \frac{1}{\beta \Delta t}, \ a_2 = \frac{1}{2\beta} - 1, \ a_3 = \frac{\gamma}{\beta \Delta t}, \ a_4 = \frac{\gamma}{\beta} - 1, \ a_5 = \frac{\Delta t}{2}\left(\frac{\gamma}{\beta} - 2\right)
\end{cases}
\tag{10}
$$

where $I$ denotes the unit matrix; and $\beta$ and $\gamma$ are the two parameters associated with integration accuracy and stability, respectively. In this paper, $\beta = 0.25$ and $\gamma = 0.50$ are adopted for the unconditionally stable integration scheme.

Without loss of generality, assume $V_0 = V(0) = 0$ and $F_0 = F(0) = 0$. Based on Equation (9), the explicit expression for the state vector of the jacket platform can be constructed in terms of the wave loading vectors at discretized time instants and can be written as follows [13]:

$$
V_i = A_{i,1}F_1 + A_{i,2}F_2 + \cdots + A_{i,i-1}F_{i-1} + A_{i,i}F_i \quad (i = 1, 2, \cdots, n)
\tag{11}
$$

where $A_{i,j}(1 \leq j \leq i \leq n)$ are the coefficient matrices, which can be expressed in the closed forms as:

$$
\begin{cases}
\mathbf{A}_{1,1} = \mathbf{Q}_2, \quad \mathbf{A}_{2,1} = \mathbf{T}\mathbf{Q}_2 + \mathbf{Q}_1, \quad \mathbf{A}_{i,1} = \mathbf{T}\mathbf{A}_{i-1,1} \ (3 \leq i \leq n) \\
\mathbf{A}_{i,j} = \mathbf{A}_{i-1,j-1} \ (2 \leq j \leq i \leq n)
\end{cases}
\tag{12}
$$

It can be seen from Equation (12) that, among the coefficient matrices, only $A_{i,1}(1 \leq i \leq n)$ needs to be calculated and stored, and the remainder can be obtained using the recurrence formula $\mathbf{A}_{i,j} = \mathbf{A}_{i-1,j-1} \ (2 \leq j \leq i \leq n)$. Therefore, the calculation and storage of the coefficient matrices involved in the explicit formulation of structural responses can be readily achieved without significant effort.

From the perspective of engineering applications, only certain critical responses of offshore structures are of interest to engineers. Suppose that $s$ is one of the critical responses. With the explicit formulation of $V_i$ in Equation (11), the explicit time-domain expression of $s$ at time instant $t_i$ can be obtained as:

$$
s_i = \boldsymbol{\phi}\, V_i = a_{i,1}F_1 + a_{i,2}F_2 + \cdots + a_{i,i-1}F_{i-1} + a_{i,i}F_i \quad (1 \leq i \leq n)
\tag{13}
$$

where:

$$
a_{i,j} = \boldsymbol{\phi}A_{i,j} \quad (1 \leq j \leq i \leq n)
\tag{14}
$$

in which $A_{i,j}(1 \leq j \leq i \leq n)$ represents the coefficient matrices shown in Equation (12); and $\boldsymbol{\phi}$ is the response transfer row vector. When $s$ is the displacement or velocity of certain DOF, $\boldsymbol{\phi}$ is a row vector consisting of 0 and 1; and when $s$ is an element force component, $\boldsymbol{\phi}$ contains entries in the element force-displacement matrix.

## 4. Random Vibration Analysis with MCS Based on ETDM

### 4.1. Digital Simulation of Wave-Particle Velocities and Accelerations

To determine the drag and incident forces using Equation (4), the wave-particle velocities and accelerations are required, which are generally modeled as zero-mean stationary Gaussian random processes. Within the framework of linear random wave theory [20], the spectral representation

method [17,21] is applied to the generation of wave-particle velocities and accelerations shown in Figure 1, which can be expressed as follows:

$$v_x(x,z,t) = \sum_{i=1}^{n_\omega} \sqrt{2\big[H_z^2(z,\omega_i)S_\eta(\omega_i)\big]\Delta\omega_i} \cdot \cos(\kappa_i x - \omega_i t + \varepsilon_i) \tag{15}$$

$$v_z(x,z,t) = \sum_{i=1}^{n_\omega} \sqrt{2\big[H'^2_z(z,\omega_i)S_\eta(\omega_i)\big]\Delta\omega_i} \cdot \sin(\kappa_i x - \omega_i t + \varepsilon_i) \tag{16}$$

$$\dot{v}_x(x,z,t) = \sum_{i=1}^{n_\omega} \sqrt{2\big[\omega_i^2 H_z^2(z,\omega_i)S_\eta(\omega_i)\big]\Delta\omega_i} \cdot \sin(\kappa_i x - \omega_i t + \varepsilon_i) \tag{17}$$

$$\dot{v}_z(x,z,t) = -\sum_{i=1}^{n_\omega} \sqrt{2\big[\omega_i^2 H'^2_z(z,\omega_i)S_\eta(\omega_i)\big]\Delta\omega_i} \cdot \cos(\kappa_i x - \omega_i t + \varepsilon_i) \tag{18}$$

where $S_\eta(\omega)$ is the power spectrum density function of the wave surface elevation; $n_\omega$ is the number of representative frequencies; $\omega_i$ $(i = 1, 2, \cdots, n_\omega)$ is the $i$-th representative frequency and $\Delta\omega_i$ is the corresponding frequency interval; $k_i$ denotes the wave number of the $i$-th cosine wave; $\varepsilon_i$ denotes the random phase angle of the $i$-th cosine wave with uniform distribution in [0, 2π]; and $H_z(z,\omega)$ and $H'_z(z,\omega)$ are the depth-dependent functions given as:

$$H_z(z,\omega) = \frac{\omega\cosh[\kappa(z+h)]}{\sinh(\kappa h)} \tag{19}$$

$$H'_z(z,\omega) = \frac{\omega\sinh[\kappa(z+h)]}{\sinh(\kappa h)} \tag{20}$$

in which $h$ is the water depth of the sea site; and $\kappa$ and $\omega$ should follow the dispersion relationship given as:

$$\omega^2 = g\kappa \cdot \tanh(\kappa h) \tag{21}$$

in which $g$ is the gravitational acceleration.

The representative frequencies in Equations (15)–(18) should be arranged at non-uniform intervals so that small frequency intervals can be achieved around the peak value of $S_\eta(\omega)$, while larger intervals can be adopted at the frequency where the value of $S_\eta(\omega)$ is small. In this regard, the representative frequencies can be obtained as follows [22]:

$$\omega_i = \frac{2\pi}{T_Z}\bigg[\pi\ln\bigg(\frac{n_\omega}{i-0.5}\bigg)\bigg]^{-\frac{1}{4}} \tag{22}$$

where $T_Z$ denotes the average zero up-crossing period of the wave surface elevation.

### 4.2. ETDM-Based MCS

Once the samples of wave-particle velocities and accelerations are generated using Equations (15)–(18), the samples of the concentrated wave loading vector *F* in Equation (8) can be obtained based on the distributed drag force vector $f_D$ and the distributed incident force vector $f_I$ shown in Equation (4). Thus far, the MCS for random vibration analysis of jacket platforms can be readily conducted based on the explicit time-domain expressions of structural responses presented in Section 3.2.

Suppose there is a total number of $M$ samples of the wave loading vector. For the $k$-th sample, $\boldsymbol{F} = \boldsymbol{F}^k(t)$, the prescribed critical response $s = s^k(t)$ at different time instants can be obtained using Equation (13), which can be expressed as:

$$s_i^k = a_{i,1}\boldsymbol{F}_1^k + a_{i,2}\boldsymbol{F}_2^k + \cdots + a_{i,i-1}\boldsymbol{F}_{i-1}^k + a_{i,i}\boldsymbol{F}_i^k \quad (1 \le i \le n;\ 1 \le k \le M) \tag{23}$$

where $s_i^k = s^k(t_i)$, $\boldsymbol{F}_j^k = \boldsymbol{F}^k(t_j)$ $(1 \le j \le i \le n)$ and $a_{i,j}$ $(1 \le j \le i \le n)$ are the coefficient row vectors shown in Equation (14).

Based on Equation (23), the mean and the variance of the critical response $s_i$ can be obtained as:

$$\mu_{si} = \frac{1}{M}\sum_{k=1}^{M} s_i^k \quad (1 \le i \le n) \tag{24}$$

$$\sigma_{si}^2 = \frac{1}{M-1}\sum_{k=1}^{M}\left(s_i^k - \mu_{si}\right)^2 \quad (1 \le i \le n) \tag{25}$$

From a statistical perspective, the mean peak values of structural responses are generally used for the design purpose of a structure. The mean peak value of the critical response $s_i$ can be obtained as:

$$s_{\text{peak}} = \frac{1}{M}\sum_{k=1}^{M} \max_{i=1}^{n}\left|s_i^k\right| \tag{26}$$

Note that the coefficient row vectors $a_{i,j}$ $(1 \le j \le i \le n)$ in Equation (23) need to be calculated just once and can be used for all of the sample analyses involved in MCS. This process has high efficiency compared with the traditional MCS, in which the equation of motion shown in Equation (8) needs to be solved for each sample of the wave loading vector. In addition, dimension-reduced MCS regarding certain critical responses of interest, rather than all the structural responses, can be easily conducted following Equation (23), which can further reduce the computational cost of MCS. To distinguish from the traditional MCS, the present approach can be termed ETDM-based MCS. For the sake of clarity, the solution procedure for the ETDM-based MCS is summarized in the flowchart presented in Figure 4.

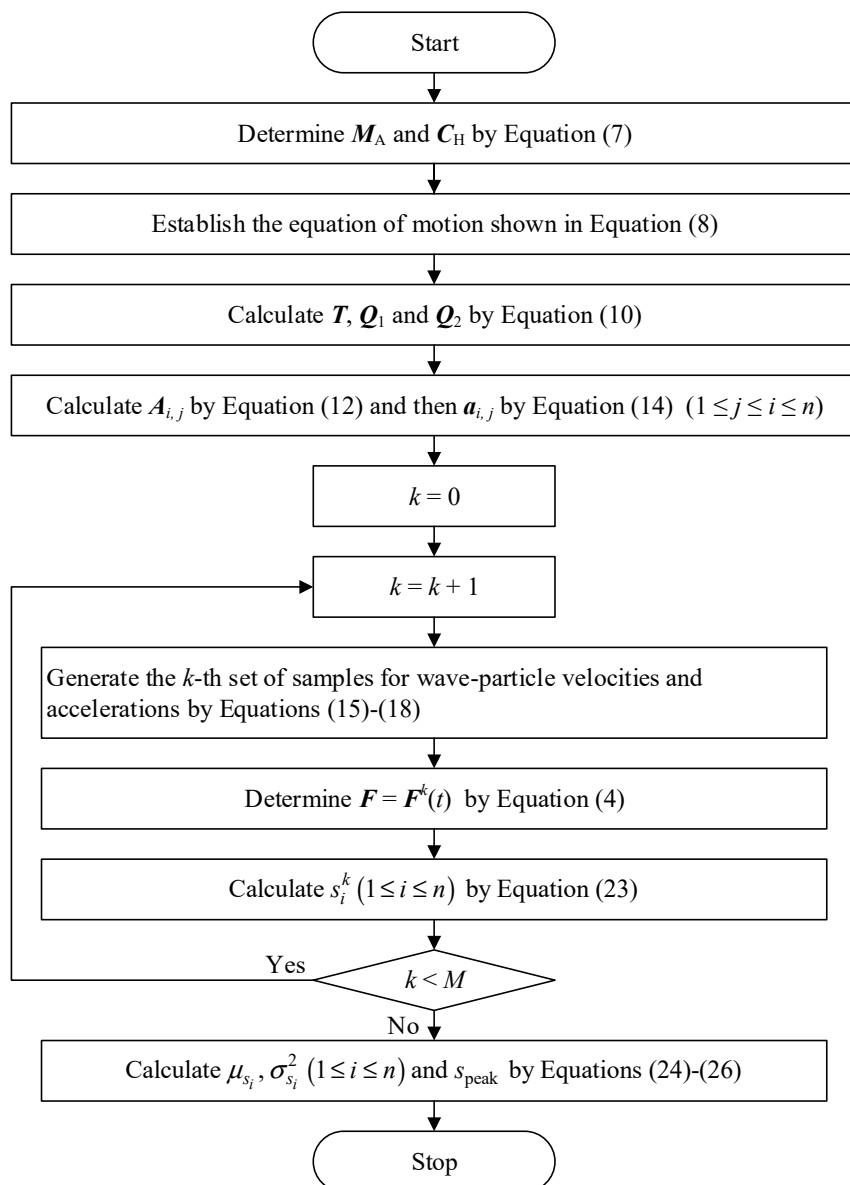

**Figure 4.** Flowchart for the explicit time-domain method (ETDM)-based Monte Carlo simulation (MCS).

## 5. Numerical Example

### 5.1. Finite Element Model

In this section, the random responses of a steel jacket platform subjected to wave loads is investigated to illustrate the accuracy and efficiency of the present approach. The jacket platform is modeled with three dimensional beam elements, and the finite element (FE) model is shown in Figure 5. The height of the jacket platform is 86.80 m, and the depth of water is 62.00 m. The pile spacing is 25.00 m in the *x*-direction and 30.00 m in the *y*-direction. The fixed ends of the piles are set at a depth of six times the pile diameter from the bottom of the jacket platform. The whole model consists of 2303 beam elements and 1964 nodes, leading to a total number of 11,688 DOFs for the entire structure. The Rayleigh damping model is applied with the structural damping ratio $\xi$ being 2% for the steel jacket platform, and the hydrodynamic damping ratio $\xi_H$ can be taken as 8% according to a recent experimental investigation [18].

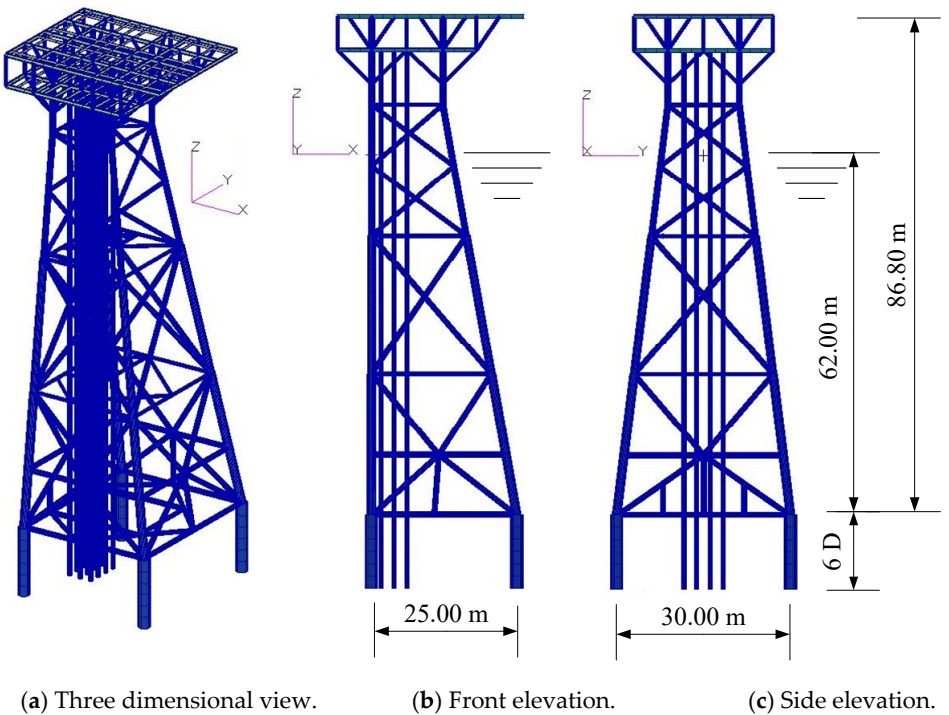

(**a**) Three dimensional view.     (**b**) Front elevation.     (**c**) Side elevation.

**Figure 5.** The finite element (FE) model of the jacket platform.

## 5.2. Wave-Particle Velocities and Accelerations

The power spectrum density function of the wave surface elevation is taken to be the two-parameter Pierson–Moskowitz wave spectrum, which is particularly suited for open sea areas and is given mathematically as:

$$S_\eta(\omega) = \frac{5}{16} \frac{\omega_P^4}{\omega^5} H_S^2 \exp\left[-\frac{5}{4}\left(\frac{\omega_P}{\omega}\right)^4\right] \tag{27}$$

where $H_S$ is the significant wave height, and $\omega_P$ is the spectral peak frequency of the wave surface elevation. For a 100-year return period, $H_S$ and $\omega_P$ are set as 11.50 m and 0.44 rad/s, respectively, and the wave spectrum is shown in Figure 6.

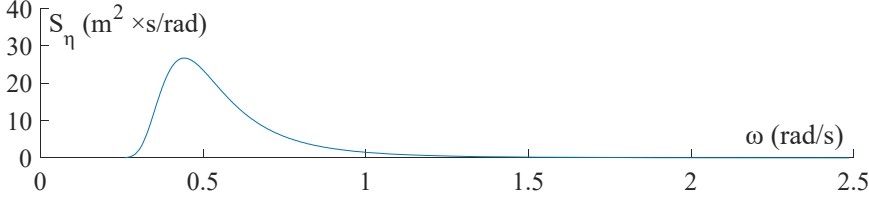

**Figure 6.** Power spectrum density function of the wave surface elevation.

The velocities and accelerations of wave particles are generated with the spectral representation method based on Equations (15)–(18), in which 400 representative frequencies are considered in the range of 0.27 to 2.30 rad/s. A set of samples of wave-particle velocities and accelerations with a duration time of $t_d = 1000$s are depicted in Figures 7 and 8, respectively. To obtain the wave loads from the wave-particle velocities and accelerations, the hydrodynamic coefficients in Equation (4) are set as $C_M = 2.00$ and $C_D = 1.30$ [6,18].

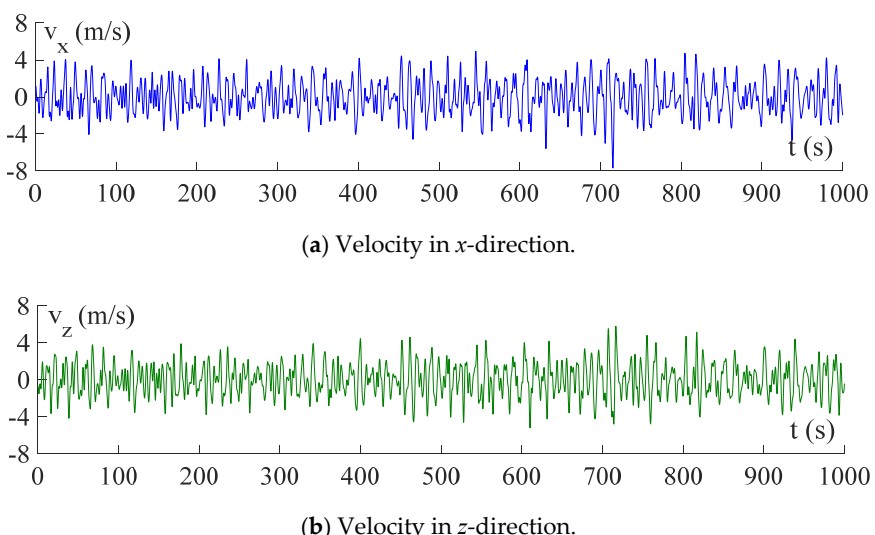

(**a**) Velocity in *x*-direction.

(**b**) Velocity in *z*-direction.

**Figure 7.** Samples of wave-particle velocities on the sea surface.

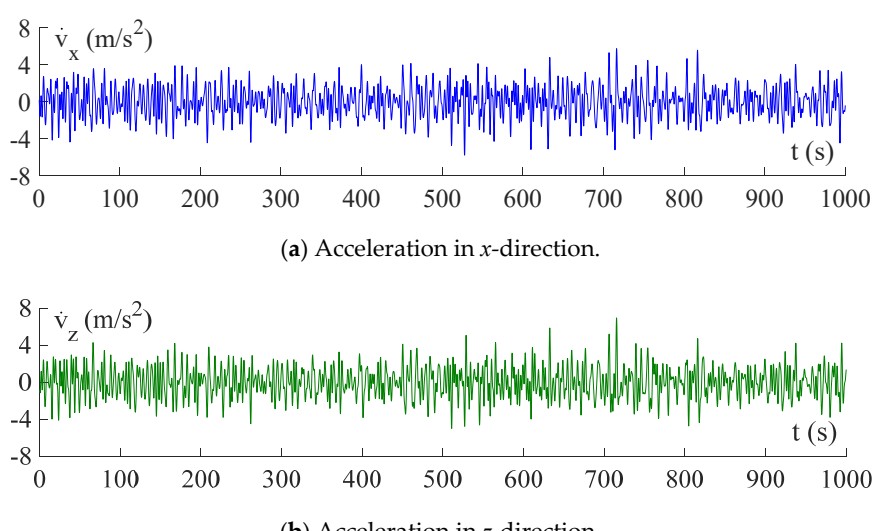

(**a**) Acceleration in *x*-direction.

(**b**) Acceleration in *z*-direction.

**Figure 8.** Samples of wave-particle accelerations on the sea surface.

*5.3. Deterministic Time-History Analysis with ETDM*

Under the action of the wave-particle velocities and accelerations shown in Figures 7 and 8, the deterministic time-history analysis of critical responses of the jacket platform is carried out with ETDM using Equation (13), in which the duration time of responses is set as $t_d$ = 1000 s with the time step set as $\Delta t$ = 0.20 s. For comparison, the time-history analysis is also conducted by solving Equation (8) directly with the Newmark-$\beta$ method, in which $t_d$ = 1000 s and $\Delta t$ = 0.20 s. The time-histories of displacement $u_x(t)$ and velocity $\dot{u}_x(t)$ at a top node of the jacket platform are shown in Figure 9a,b respectively, and the time-histories of bending stress $\sigma(t)$ and shear stress $\tau(t)$ at the bottom of a battered leg are shown in Figure 10a,b, respectively. It can be seen from the above figures that the results obtained with ETDM are identical to those obtained with the Newmark-$\beta$ method, indicating the correctness of the explicit formulation of structural responses in ETDM.

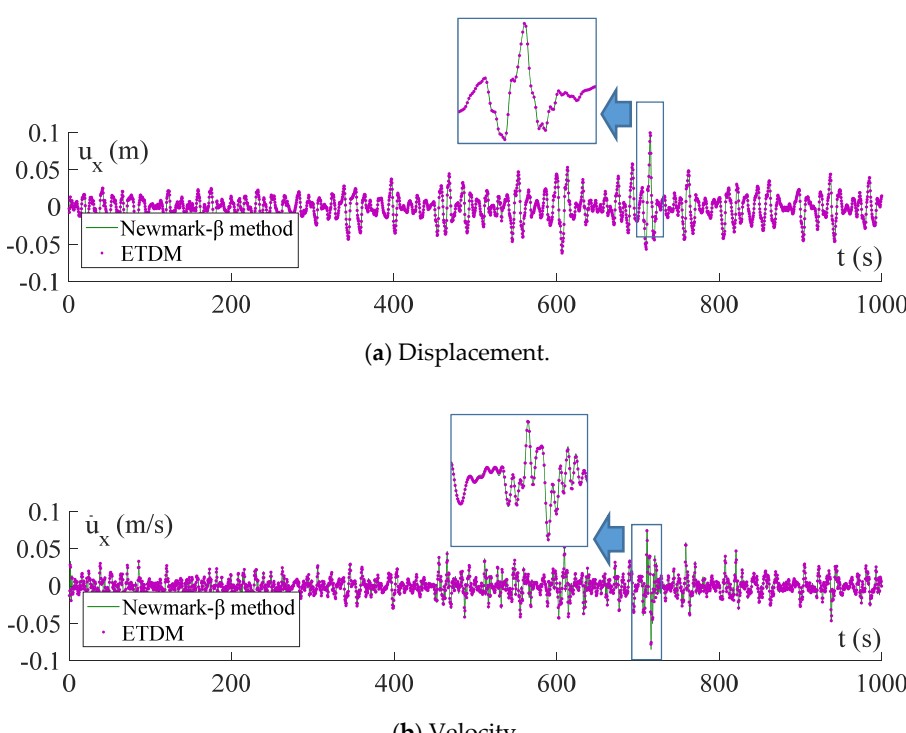

(**a**) Displacement.

(**b**) Velocity.

**Figure 9.** Displacement and velocity at a top node of the jacket platform.

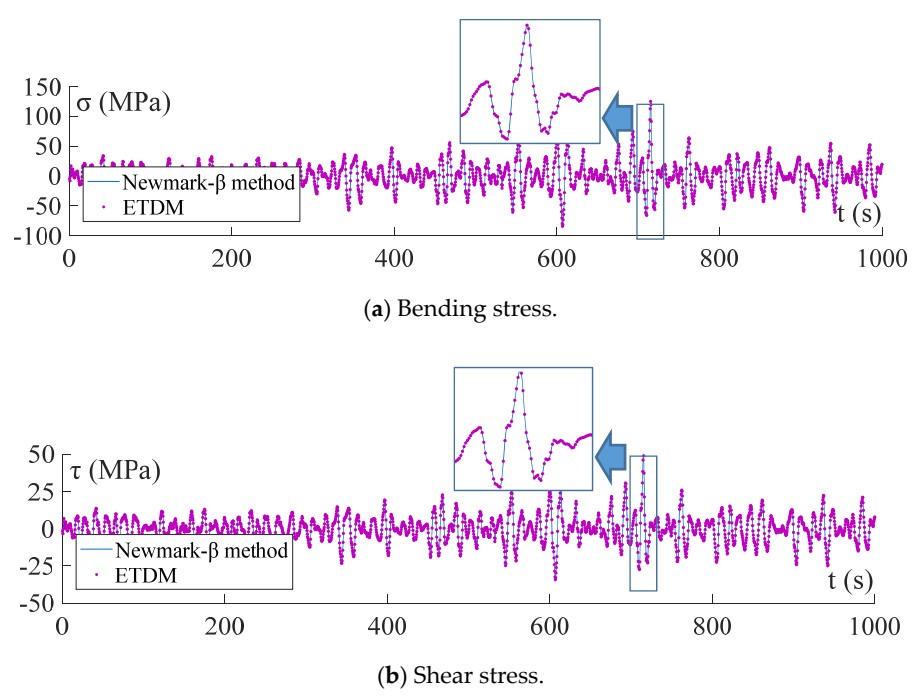

(**a**) Bending stress.

(**b**) Shear stress.

**Figure 10.** Bending and shear stress at the bottom of a battered leg of the jacket platform.

*5.4. Random Vibration Analysis with ETDM-Based MCS*

The ETDM-based MCS stated in Section 4.2 is now used to conduct the random vibration analysis of the jacket platform with the nonlinear Morison equation, in which the samples of wave-particle velocities and accelerations are generated in Section 5.2 with a sample size of $M = 1000$. For the purpose of comparison, the PSM with the linearized Morison equation [9,10] is also used for the analysis of

standard deviations of structural responses, and the mean peak value of a critical response is further obtained using the following equation [12]:

$$s_{\text{peak}} = \left( \sqrt{2 \ln N_s} + \frac{0.57716}{\sqrt{2 \ln N_s}} \right) \sigma_s \qquad (28)$$

where $\sigma_s$ is the standard deviation of the critical response; and $N_s$ denotes the number of wave cycles within the duration time under the consideration.

To investigate the influence of the linearization of the Morison equation on the response statistics of the jacket platform, different significant wave heights and spectral peak frequencies are considered in the above statistical analysis. The significant wave height $H_S$ is taken as 3.25, 5.00, 7.50, and 11.50 m, respectively, and the corresponding spectral peak frequency $\omega_P$ is taken as 0.76, 0.64, 0.53, and 0.44 rad/s, respectively. The standard deviations of displacement $u_x(t)$ and velocity $\dot{u}_x(t)$ at a top node of the jacket platform are shown in Figure 11a,b, respectively, and the standard deviations of bending stress $\sigma(t)$ and shear stress $\tau(t)$ at the bottom of a battered leg are shown in Figure 12a,b, respectively. The mean peak values of the above four critical responses are depicted in Figures 13 and 14. It can be observed from Figures 11 and 12 that the standard deviations of displacement and stress responses obtained with the two methods are in good agreement, while for the standard deviations of the velocity response, the discrepancy between the results of the two methods reaches up to 16%. Furthermore, in Figures 13 and 14, with the increase in significant wave height, large discrepancies up to 94% can be found between the mean peak values of responses obtained with the two methods. The mean peak values are underestimated to a large extent by the method based on the linearized Morison equation for high sea conditions with dominating wave drags. This phenomenon is consistent with the observation reported in the literature [11,12].

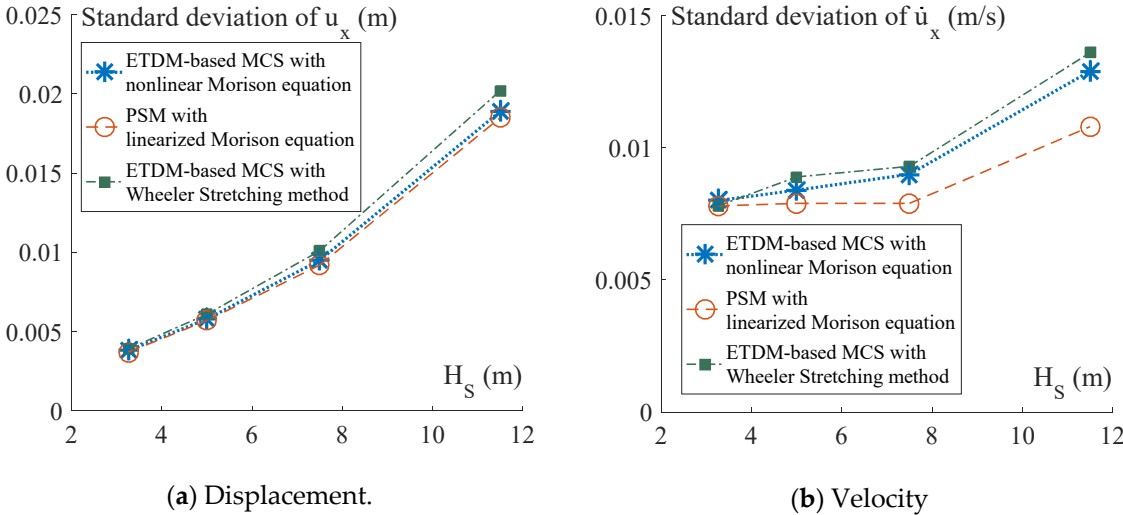

(a) Displacement.　　　　　　　　　　　　　　　(b) Velocity

**Figure 11.** Standard deviations of the displacement and velocity at a top node of the jacket platform.

To further consider the influence on the jacket platform of the wave loads above the still water level, the Wheeler stretching method is used to modify the profiles of the wave-particle velocities and accelerations [23,24], and the wave loads are modified accordingly using Equation (4). The corresponding standard deviations and mean peak values of the four critical responses are obtained with ETDM-based MCS, which are also depicted in Figures 11–14 for comparison. It can be observed from Figures 11–14 that with the increase in the significant wave height, an increase up to 7% occurs in both standard deviations and mean peak values of the critical responses when the Wheeler stretching method is used to account for the effects of the wave loads above the still water level.

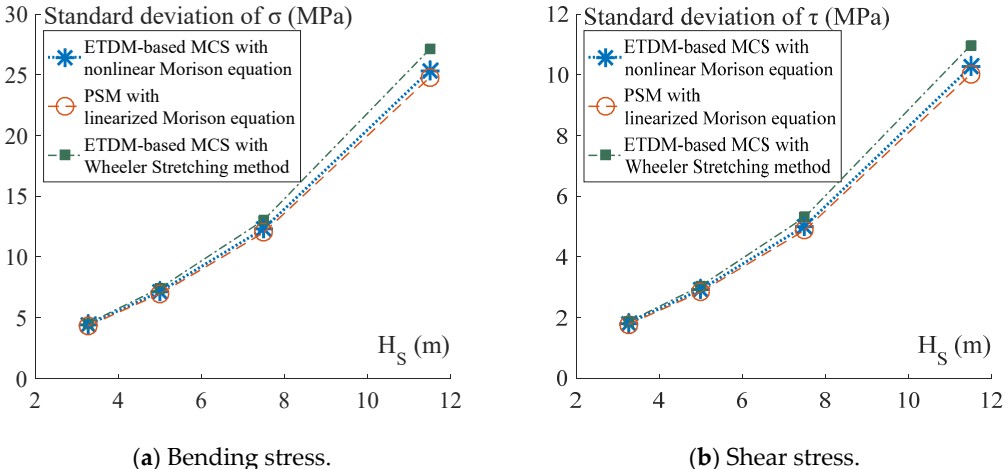

(**a**) Bending stress.　　　　　　　　　　(**b**) Shear stress.

**Figure 12.** Standard deviations of the bending and shear stress at the bottom of a battered leg of the jacket platform.

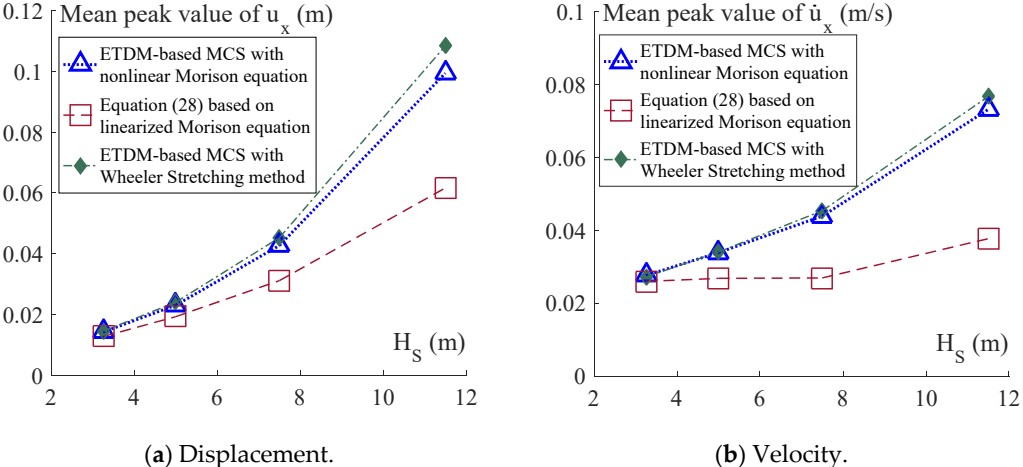

(**a**) Displacement.　　　　　　　　　　(**b**) Velocity.

**Figure 13.** Mean peak values of the displacement and velocity at a top node of the jacket platform.

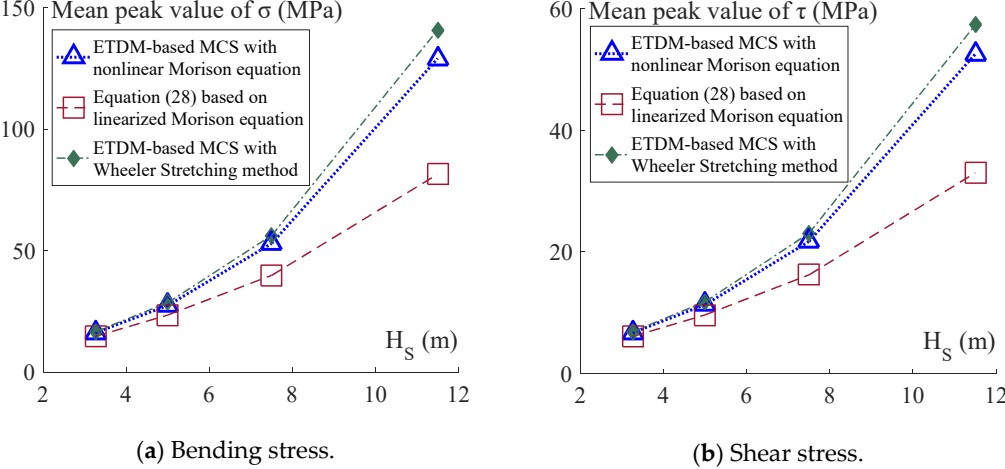

(**a**) Bending stress.　　　　　　　　　　(**b**) Shear stress.

**Figure 14.** Mean peak values of the bending and shear stress at the bottom of a battered leg of the jacket platform.

To investigate the computational efficiency, the time elapsed by ETDM-based MCS and the traditional MCS is presented in Table 1. It takes about 200 s to calculate the coefficient matrices for construction of the explicit expressions of dynamic responses, and then only a total time of 100 s is required for MCS with 1000 sample analyses based on the explicit formulation of responses. In contrast, the computation time for the traditional MCS reaches $10^6$ s with the same sample size, in which the equation of motion of the structure needs to be solved for each sample analysis. It can be seen that the present ETDM-based MCS has high efficiency and can be effectively used for the time-domain random vibration analysis of jacket platforms under wave loads, in which the nonlinear Morison equation for the wave drags can be readily taken in consideration.

**Table 1.** Computation time of ETDM-based MCS and the traditional MCS with 1000 sample analyses.

| Method | Elapsed Time (s) |
| --- | --- |
| ETDM-based MCS | 200 + 100 = 300 |
| Traditional MCS | $10^6$ |

Note: all the above computations were performed on a PC with an Intel Core i7-3630QM CPU@2.40 GHz processor and 24 GB RAM.

## 6. Conclusions

For jacket platforms under random wave loads, the time-domain MCS is considered the optimal choice for consideration of the nonlinear Morison equation of wave drags, and for evaluation of the mean peak values of stochastic responses without additional assumptions regarding the probability distribution of responses. However, the large computational cost obstructs the application of the traditional MCS to engineering practice. ETDM with explicit expressions of dynamic responses addresses the bottleneck of the extremely low efficiency of MCS and makes it feasible for practical application to engineering problems. In this paper, the ETDM-based MCS was applied to the random vibration analysis of a jacket platform with 11,688 DOFs subjected to random wave loads, indicating the high accuracy of the present approach compared with the method based on the linearized Morison equation, and the high efficiency compared with the traditional MCS.

**Author Contributions:** C.S. conceived the overall approach and the main conceptual design of the article. W.L. implemented the approach and drafted the manuscript. Y.T. helped with revising the content of the text. All authors have read and agreed to the published version of the manuscript.

**Funding:** This research was funded by the National Natural Science Foundation of China, grant number 51678252, and the Science and Technology Program of Guangzhou, China, grant number 201804020069.

**Conflicts of Interest:** The authors declare that they have no conflict of interest.

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
