# Peer review of "Explicit Time-Domain Approach for Random Vibration Analysis of Jacket Platforms Subjected to Wave Loads"

_jmse, doi:10.3390/jmse8121001_

Round 1

Reviewer 1 Report

Dear Authors,

Congratulations to the authors. In my opinion the paper should be published in current form.

Reviewer 2 Report

The authors have identified the effect of taking the wave crest load into account by using the Wheeler stretching method.

The paper is much improved by this change

As the main objective of the paper is to study the Explicit time-domain approach for random vibration, the paper is now in acceptable format.

This manuscript is a resubmission of an earlier submission. The following is a list of the peer review reports and author responses from that submission.

Round 1

Reviewer 1 Report

This well written paper presents an interesting novel application of  an explicit time-domain approach to vibration problems. The method is clearly explained in detail and validated against a more widely used alternative approach.

I can only provide a few suggestions which the authors might want to consider:

Line 42: "dominant" instead of major?

Line 138: Move the references [14-16] up right behind the words integration scheme in the line above

Which FEA Code did you use for your implementation?

To clarify the differences between the two methods, you might provide a scheme similar to Figure 4 for the conventional MCS.

Why did you choose a relatively complex complete platform model for validation? I would always rather try to test a new numerical tool against the simplest possible experiment/analytical solution.   

Reviewer 3 Report

The paper presents a comparison between ETDM based MSC and traditional MSC, in principle very good comparison is obtained.

The paper uses forces on jackets as example. The problem is, however, that the calculation of the forces on the jacket is represented by a linear kinematics model and there is no discussion on how the forces above the still water line (in the wave crest) are accounted for. Linear Airy wave theory is not applicable in the crest and there is no discussion regarding this. So the force calculations cannot be trusted.

Furthermore, a discussion of how to obtain design forces from the mean values and standard deiations is required in the paper.

Round 2

Reviewer 2 Report

Thank you for your reply.

Reviewer 3 Report

The authors have answered to my questions, however, the answers are not followed up with any changes in the manuscript as I see it.

I request a discussion of the calculation of wave forces and associated contribution to the overturning moment from the wave above the still water level.

I also request that the design wave load and the overturning moment be calculated. It is not sufficient to state that equation (28) apply. Is the mean peak value same as the caracteristic design value?

I want the points to be discussed in the paper